# Spatially periodic computation in the entorhinal-hippocampal circuit during navigation

**Bo Zhang[1,2,3], Xin Guan[2], Dean Mobbs[3], Jia Liu[2]\***

[1]Beijing Academy of Artificial Intelligence, Beijing, China; [2]Tsinghua Laboratory of Brain and Intelligence & Department of Psychological and Cognitive Sciences, Tsinghua University, Beijing, China; [3]Division of Humanities and Social Sciences, California Institute of Technology, Pasadena, United States

## eLife Assessment

This study offers **important** insights into how entorhinal and hippocampal activity support human thinking in feature spaces. It replicates hexagonal symmetry in entorhinal cortex, reports a novel threefold symmetry in both behavior and hippocampal signals, and links these findings with a computational model. The task and analyses are sophisticated, and the results appear **convincing** and of broad interest to neuroscientists.

**\*For correspondence:**
liujiaTHU@tsinghua.edu.cn

**Competing interest:** The authors declare that no competing interests exist.

**Abstract** To enable navigation in both physical and mental spaces, the human brain employs a cognitive map constructed from the global metrics of the entorhinal cortex and the local representations of the hippocampus. However, how these two regions coordinate to enable navigation remains poorly understood. Here, we designed an object-matching task where human participants unknowingly manipulated object variants arranged in a ring-like structure around a central prototype. Functional MRI revealed a threefold spatial periodicity in the hippocampal activity that tracked navigation directions from object variants to the central prototype. This hippocampal periodicity was phase-locked with the well-documented sixfold periodicity of the entorhinal cortex, suggesting hierarchical interaction between these regions. Consistent with this neural pattern, a corresponding threefold periodicity was observed in behavioral performance, which was synchronized with hippocampal activity. Finally, an EC-HPC PhaseSync model reproduced this phenomenon, in which the sixfold activity periodicity of entorhinal grid cells across directions projects vectorial representations to the hippocampus, and the collection of these vectors exhibits threefold periodicity to represent conceptual directions. Together, these findings reveal a periodic mechanism through which entorhinal grid codes structure hippocampal vector representations.

## Introduction

The cognitive map, initially introduced by *Tolman, 1948* and later supported by the discovery of place cells in the hippocampus (HPC; *O'Keefe and Dostrovsky, 1971*), has served as a framework for spatial navigation. Grid cells in the upstream entorhinal cortex (EC), known for their hexagonal firing pattern to provide the metric for space (*Hafting et al., 2005*), have also been found to represent conceptual space beyond the physical reference frame (*Constantinescu et al., 2016*; *Bao et al., 2019*; *Raithel et al., 2023*). These findings suggest that the EC-HPC circuit fundamentally organizes spatial and non-spatial knowledge (*Epstein et al., 2017*; *Behrens et al., 2018*; *Bottini and Doeller, 2020*; *Park*

*et al., 2020*), and to guide flexible behaviors such as retrieving knowledge from the past and making decisions for the future (*Addis et al., 2007*; *Hassabis and Maguire, 2007*; *Schacter et al., 2012*).

Flexible navigation requires the mental simulation of prospective pathways on a cognitive map, which is defined by essential cues such as self-location, goal location, direction, and distance (*Nyberg et al., 2022*). This process necessarily involves dynamic representations of the external world relative to self-location. Converging evidence indicates that the HPC binds spatial cues into vectorial representations, providing activity gradients that reflect potential pathways toward goals. In bats, a subpopulation of CA1 neurons exhibited conjunctive tuning to both direction and distance, with activity gradually decaying within 10–15 min after the goal is displaced (*Sarel et al., 2017*). In rats, similar findings were observed for neurons tuned to featureless goal locations distributed across space (*Ormond and O'Keefe, 2022*). These neurons fire maximally when the animal is oriented toward the goal and rapidly reorganize following goal shifts. Consistently, the neural representation of prospective pathways in the human brain is affected by goal locations (*Muhle-Karbe et al., 2023*). The neural geometry derived from the BOLD signals in the HPC is distorted by goals, reflecting the successful learning of goal-directed navigation. Collectively, these findings suggest that the formation of prospective pathways relies on memory-based vectorial representations in the HPC. However, these studies raise the question of how grid cells in the EC contribute to this process.

We hypothesize that projections from EC grid cell populations provide a coherent cognitive-map framework in the HPC that embeds a threefold periodic structure across spatial directions to support vectorial representations and simulating the prospective pathways. Grid cells exhibit hexagonal firing patterns with nearly invariant orientations (*Hafting et al., 2005*; *Sargolini et al., 2006*; *Krupic et al., 2012*; *Gardner et al., 2022*). These properties may support the formation of vector-like representations of pathways by coactivation of grid cell populations. In the simplest case, when one mentally simulates a straight pathway aligned with the grid orientation, a subpopulation of grid cells would be sequentially activated, and the resulting population activity would manifest a near-perfect vectorial representation with constant activity strength along the pathway. In contrast, when the pathway is misaligned with the grid orientation, the corresponding grid cell population yields a distorted vectorial code. Consequently, simulating straight pathways spanning 0°–360° yields only half the number of unique activity patterns. This arises because the hexagonal grid's 180° rotational symmetry makes orientations separated by 180° indistinguishable. We therefore speculate that the vectorial representations embedded in grid cell activity are periodic across spatial orientations and are transmitted through the EC–HPC circuit to bind prospective path directions. Consistent with this idea, reorientation paradigms in both rodents and young children demonstrate that subjects search equally at two opposite directions, reflecting successful orientation encoding but a failure to integrate spatial direction (*Hermer and Spelke, 1994*; *Julian et al., 2015*; *Gallistel, 2017*; *Julian et al., 2018*).

This hypothesis is supported by evidence from anatomical, functional, computational, and physiological findings on the EC–HPC circuit. Anatomically, the EC serves as one of the major sources of input to the downstream HPC (*Witter and Amaral, 1991*; *van Groen et al., 2003*; *Garcia and Buffalo, 2020*). Functionally, grid cells in the medial EC exhibited multiplexed and heterogeneous responses corresponding to position, direction, and speed before they are integrated in the HPC (*Sargolini et al., 2006*; *Hardcastle et al., 2017*). Computationally, grid cells have been proposed as the foundation of hippocampal place field formation by integrating multiple grid modules (*Solstad et al., 2006*; *de Almeida et al., 2009*; *Bush et al., 2014*; *Bush et al., 2015*; *Bicanski and Burgess, 2019*). Physiologically, EC lesions disrupt the precision and stability of place fields (*Hales et al., 2014*), leading to reduced discharge rates and field sizes (*Van Cauter et al., 2008*). If our hypothesis is correct, a threefold periodicity aligned with the three principal grid axes should emerge in the HPC, phase-locked with EC activity along path directions. Because simultaneous population-level recordings from the EC and HPC remain technically challenging, we employed fMRI, which has previously revealed sixfold periodicity in the EC (*Doeller et al., 2010*; *Constantinescu et al., 2016*; *Bao et al., 2019*; *Wagner et al., 2023*; *Raithel et al., 2023*), to test for periodicity in the HPC.

A novel 3D object, named Greeble (*Figure 1A*; *Gauthier and Tarr, 1997*), was used to create a conceptual Greeble space. Within this space, locations were represented by Greeble variants characterized by two features ('Loogit' and 'Vacso'). The feature length defined the two dimensions of the space. The central Greeble served as the prototype. Participants were instructed to morph Greeble variants to match this target prototype (*Figure 1B*). This process generated a sequence

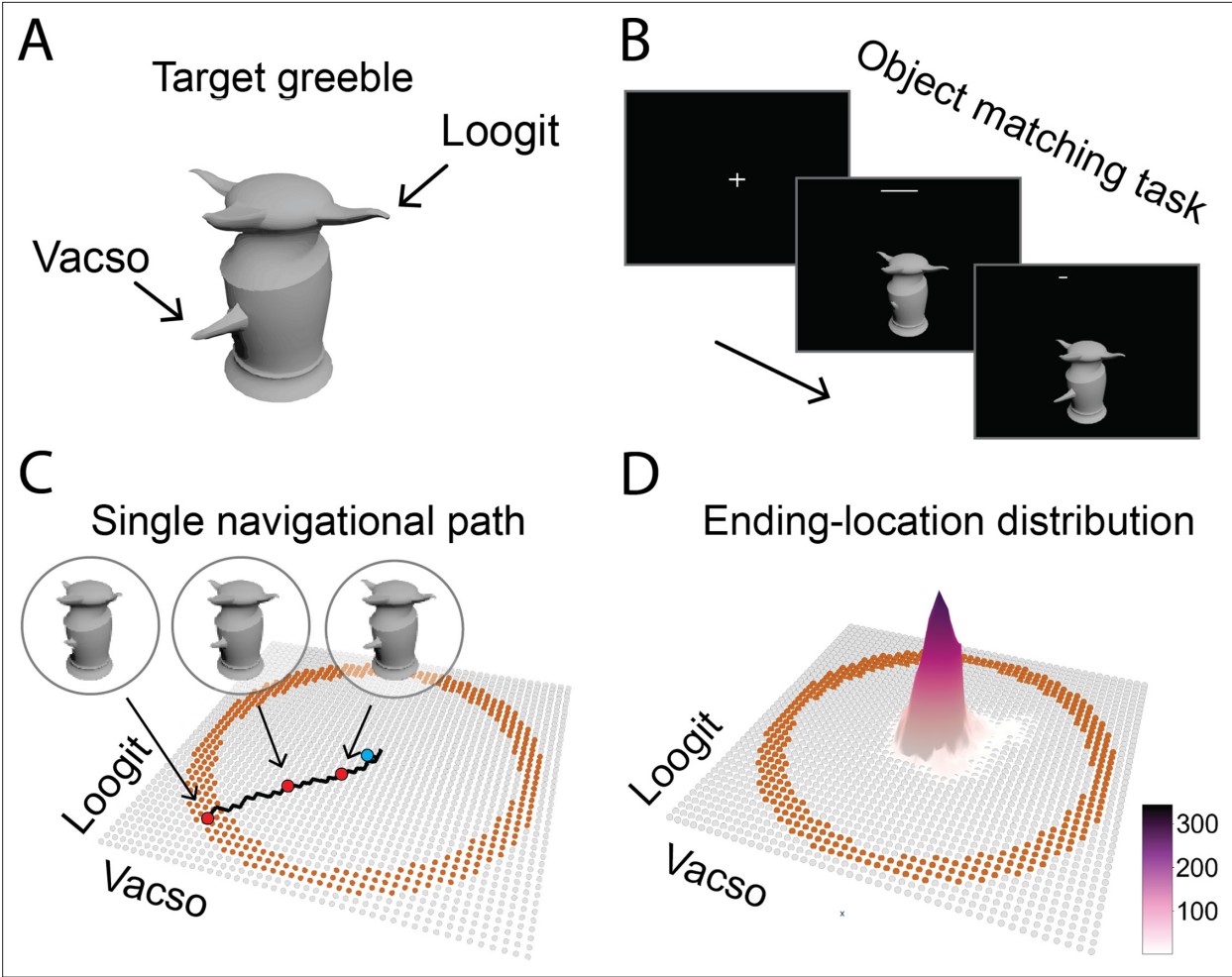

**Figure 1.** Experimental design. (**A**) Depiction of the Greeble prototype (*Gauthier and Tarr, 1997*) and its two defining features, namely 'Loogit' and 'Vacso'. (**B**) Inside the MRI scanner, participants adjusted the length of Loogit and Vacso to match the prototype by stepwise button presses, within a 10 s time limit. (**C**) Conceptual object space. Each orange dot within the ring-shaped area represents a Greeble variant, while the central blue dot indicates the Greeble prototype (i.e. the goal location). The red dots denote exemplar intermediate locations along the navigational path (i.e., the black line). (**D**) Density distribution of participants' ending locations indicated an overall superior behavioral performance for detecting the periodic activity of the HPC.

The online version of this article includes the following source data and figure supplement(s) for figure 1:

**Source data 1.** Related to *Figure 1D*.

**Figure supplement 1.** The averaged number of path directions across all paths (i.e., trials).

**Figure supplement 2.** Histograms of path directions for each participant.

**Figure supplement 3.** Learning effect of the object matching task.

**Figure supplement 4.** Movement paths of human participants during Greeble morphing in the MRI experiment.

of Greebles that resembled movements along navigational path in a two-dimensional conceptual space (*Figure 1C*), although participants were unaware of the underlying Greeble space. To ensure a comprehensive exploration of the Greeble space for detecting hippocampal periodicity, Greeble variants were pseudo-randomly sampled at the periphery of space. This ensured a high-resolution sampling of conceptual directions ranging from 0° to 360° (i.e. the orange locations in *Figure 1C*), while controlling for distance. As a result, we observed a threefold periodicity in hippocampal activity, cross-validated using sinusoidal modulation and spectral analyses. The spatial phase of the HPC was coupled with the sixfold periodicity in the EC; no spatial offset was identified. In addition, we identified a threefold periodicity in participants' behavioral performance that was phase-locked with hippocampal activity. Finally, the EC–HPC PhaseSync model, developed to simulate EC projections into the

HPC, reproduced the emergence of threefold activity periodicity across directions under randomized goal locations. Together, these empirical findings highlight a periodic representation of conceptual directions within the HPC, suggesting that vectorial representations in the hippocampus may arise through projections from periodic grid codes in the EC.

## Results

### Behavioral results

Participants adjusted Greeble features using two response boxes inside the MRI scanner. Each response box controlled one feature, with two buttons used to stretch and shrink the feature, respectively. To prevent a 'horizontal–vertical movement' strategy, in which participants navigate primarily along the cardinal directions (e.g. repeatedly adjusting one feature towards North and then East), participants were encouraged to adjust both features simultaneously to promote directional variability in path directions for detecting neural periodicity. This strategy, referred to as 'Radial Adjustment', involved making a one-unit adjustment to one feature with the left hand followed by a one-unit adjustment to the other feature with the right hand.

To examine participants' movement strategy, we first calculated the number of unique directions across every three consecutive steps within their original movement paths (*Figure 1—figure supplement 1*). Participants showed an average of 3.8 directions per path, which was significantly higher than the two directions predicted under the 'horizontal–vertical movement' strategy (t(32) = 15.76, p<0.001, two-tailed; Cohen's d=2.78). Next, we examined the uniformity of directional distributions across all generated paths spanning 0°–360°. No significant deviation from uniformity was observed (*Figure 1—figure supplement 2*; p>0.05, Rayleigh test for circular uniformity). These results confirm that the 'Radial Adjustment' strategy was stably adopted by participants during navigation in the conceptual Greeble space, thereby reducing potential bias in subsequent analyses.

To eliminate learning effects on BOLD signal periodicity, all participants completed a training task one day prior to the MRI scanning to familiarize themselves with Greebles morphing (see Materials and methods for details). Behavioral performance was defined as a composite score combining path length and error size, with superior performance characterized by smaller scores. Learning effects were observed during the practice experiment on day 1 (*Figure 1—figure supplement 3*; t(32) = –2.46, p=0.019, two-tailed; Cohen's d=0.44), but were no longer present during the MRI experiment on day 2 (t(32) = –0.74, p=0.462, two-tailed; Cohen's d=0.13). These results suggest that participants maintained stable task accuracy across experimental sessions during MRI scanning. (*Figure 1D*; Paths of individual participants are shown in *Figure 1—figure supplement 4*).

### Sixfold periodicity in the EC

Using sinusoidal modulation, the grid orientation, reflecting the allocentric direction of the grid axes, was calculated for each voxel within the bilateral EC using half of the dataset (experimental sessions 1, 3, 5, and 7). Participants' path directions were determined by the 2D position of Greeble variants relative to the ending locations, with 0° arbitrarily defined as movement from the East (*Figure 2A*). Significant deviations from uniformity were observed in 30 out of 33 participants (*Figure 2—figure supplement 1*, p<0.05; Rayleigh test of uniformity and pairwise phase consistency (PPC), Bonferroni-corrected across participants), indicating that voxelwise grid orientations in the EC were consistent for the majority of participants.

Hexagonal sixfold activity was examined using the other half of the dataset (experimental sessions 2, 4, 6, and 8), with path directions calibrated according to participant-dependent grid orientations. This analysis identified a significant cluster in the right EC (*Figure 2B*; initial threshold: p=0.05, two-tailed; cluster-based SVC correction for multiple comparisons: p<0.05; Cohen's d=0.63; Peak MNI coordinate: 32, –6, –30). Four ROI-based control analyses were performed using rotationally symmetric parameters (three-, four-, five-, and sevenfold) to validate the robustness of the cluster in representing the sixfold periodicity. The ROI was defined as a functional mask of the right EC identified in the voxel-based analysis and further restricted within the anatomical EC. These analyses revealed significant periodic modulation only at sixfold (*Figure 2C*; t(32) = 3.56, p=0.006, two-tailed, corrected for multiple comparisons across rotational symmetries; Cohen's d=0.62). In contrast, no significant effects were observed for threefold periodicity (120° periodicity; t(32) = 1.10, p=0.28, Cohen's d=0.19),

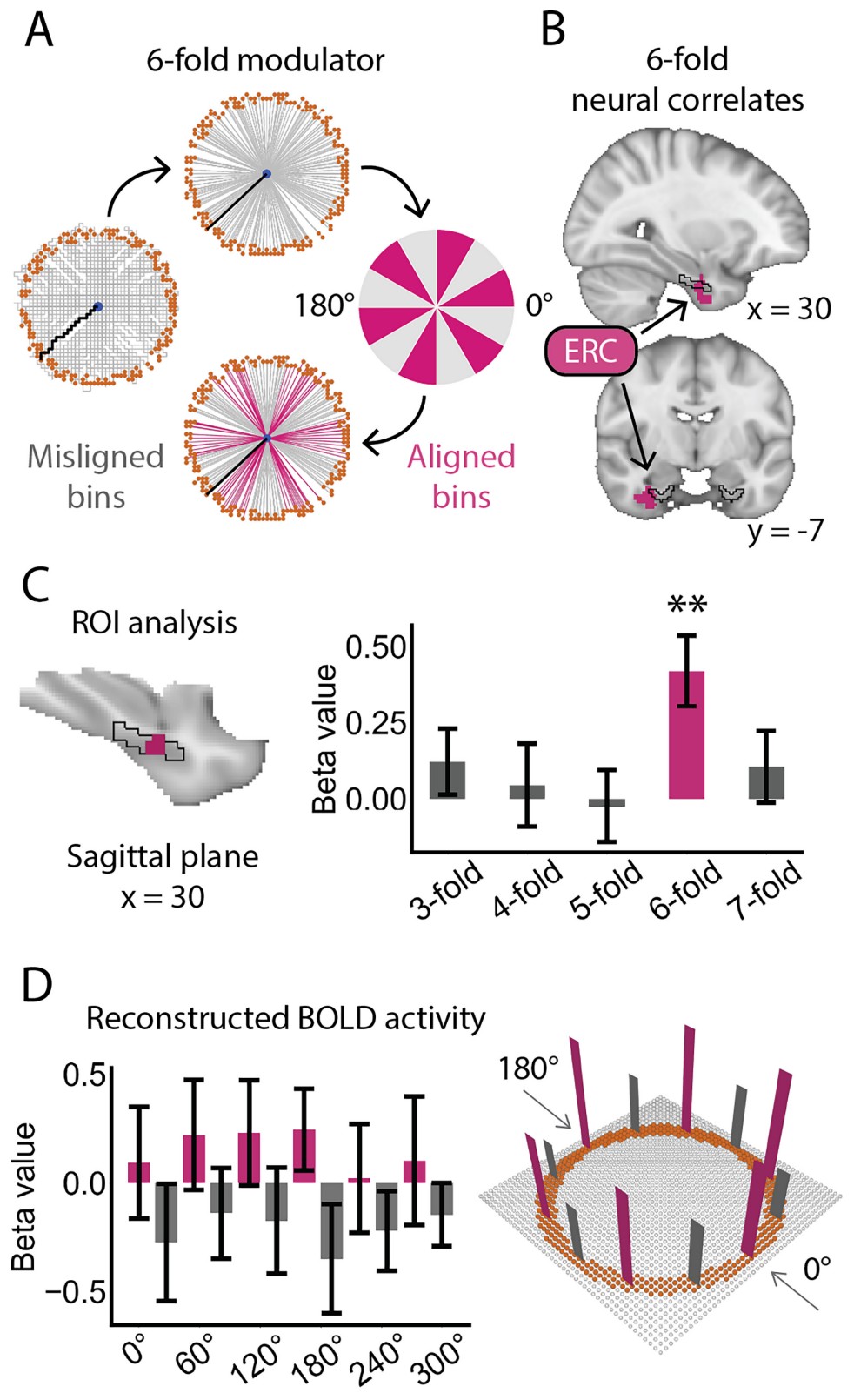

**Figure 2.** Sixfold periodicity in the EC. (**A**) Schematic defining path directions in the sixfold modulation.
Participants' path directions (Top) were extracted from the original paths (Left) by connecting the starting
and ending locations, with 0° arbitrarily set to the East as the reference, and were then classified as 'aligned'
or 'misaligned' (right and bottom). The original paths of each participant were referred to *Figure 1—figure*

*Figure 2 continued on next page*

*Figure 2 continued*

**supplement 4**. (**B**) Voxel-based sinusoidal modulation revealed significant sixfold periodicity within the right EC (voxel-based analysis: initial threshold: p=0.05, two-tailed; cluster-based small volume correction (SVC) for multiple comparisons: p<0.05; Cohen's d=0.63; Peak MNI coordinate: 32, –6, –30). Volumetric results are displayed in radiological orientation; numbers below the brain slices indicate MNI coordinates. (**C**) ROI-based analysis, using a functional mask derived from the significant right EC cluster and constrained within the anatomical EC, confirmed a significant sixfold periodicity (t(32) = 3.56, p=0.006, two-tailed, corrected for multiple comparisons across rotational symmetries; Cohen's d=0.62). The black line indicates the boundary of the EC. (**D**) Schematic illustrating the sixfold directional tuning curve reconstructed from sinusoidal modulation (Left) and its representation in the 2D Greeble space (Right). The error bars indicate SEM.

The online version of this article includes the following source data and figure supplement(s) for figure 2:

**Source data 1.** Related to *Figure 2C*.

**Source data 2.** Related to *Figure 2D*.

**Figure supplement 1.** Distribution of grid orientations in the EC.

---

fourfold periodicity (90° periodicity; t(32) = 0.31, p=0.76, Cohen's d=0.05), fivefold periodicity (72° periodicity; t(32) = –0.21, p=0.83, Cohen's d=–0.04), or sevenfold periodicity (51.4° periodicity; t(32) = 0.88, p=0.39, Cohen's d=0.16).

We further examined the fluctuation of hexagonal activity across the angular domain of path directions, following the approach reported in previous studies (*Doeller et al., 2010*; *Constantinescu et al., 2016*; *Bao et al., 2019*; *Wagner et al., 2023*; *Raithel et al., 2023*). In these studies, EC activity exhibited six peaks as a function of path direction. To do so, we reconstructed the BOLD signals using sinusoidal modulation for each path direction and each participant. To account for individual differences in spatial phase, path directions were calibrated by subtracting the participant-specific grid orientation estimated from the odd sessions. The directional tuning curve was then derived by averaging the reconstructed BOLD signals across voxels within a hand-drawn bilateral EC mask and across participants. A stable sixfold periodicity was observed (*Figure 2D*, left; p<0.05, permutation corrected for multiple comparisons), characterized by stronger activity when path directions were aligned with the calibrated grid axes and weaker activity when they were misaligned, confirming a 60° periodicity of EC activity across conceptual directions in the Greeble space (*Figure 2D*, right).

## Threefold periodicity in the HPC

Potential neural periodicity in the HPC was assessed using a Fast Fourier Transform (FFT)-based spectral analysis. FFT was employed to permit unbiased identification of multiple candidate periodicities (e.g. from three to sevenfold), independent of any a priori assumptions regarding spatial phase (orientation). Unlike the EC, where a sixfold periodicity is well established, the periodic structure of the HPC was not known a priori. FFT was thus applied as a phase-invariant approach to detect potential periodicities. Moreover, FFT was employed as an independent analysis to cross-validate the sinusoidal modulation results, thereby offering complementary evidence for the sixfold periodicity in EC and the threefold periodicity in HPC. Accordingly, BOLD signals from all eight experimental sessions were modeled using a GLM to estimate direction-dependent activity. Participants' path directions were down-sampled into 10° bins, resulting in a total of 36 directional bins (e.g. 0°, 10°, 20°, etc.). These bins were entered into the GLM as binary regressors. The resulting direction-dependent parametric maps were sorted in ascending order of spatial directions, and FFT was applied to extract spectral magnitude maps for periodicities ranging from three to sevenfold (*Figure 3A*; see Materials and methods for details).

A significant cluster of threefold periodicity was observed in the bilateral HPC (*Figure 3B*; initial threshold: p=0.05, two-tailed; Cluster-based SVC corrected for multiple comparisons: p<0.05; Cohen's d=1.06; Peak MNI coordinate: –24, –20, –18). Moreover, a significant cluster of sixfold periodicity was identified in the right EC (*Figure 3C*; initial threshold: p=0.05, two-tailed; Cluster-based SVC-corrected for multiple comparisons: p<0.05; Cohen's d=1.27; Peak MNI coordinate: –22, –14, –30), thereby replicating the finding from the sinusoidal modulation. No significant periodic activity was observed for the other folds (p>0.05).

The hippocampal threefold periodicity has not been previously reported in the spatial domain by either neurophysiological or fMRI studies. To validate its reliability, we conducted additional analyses.

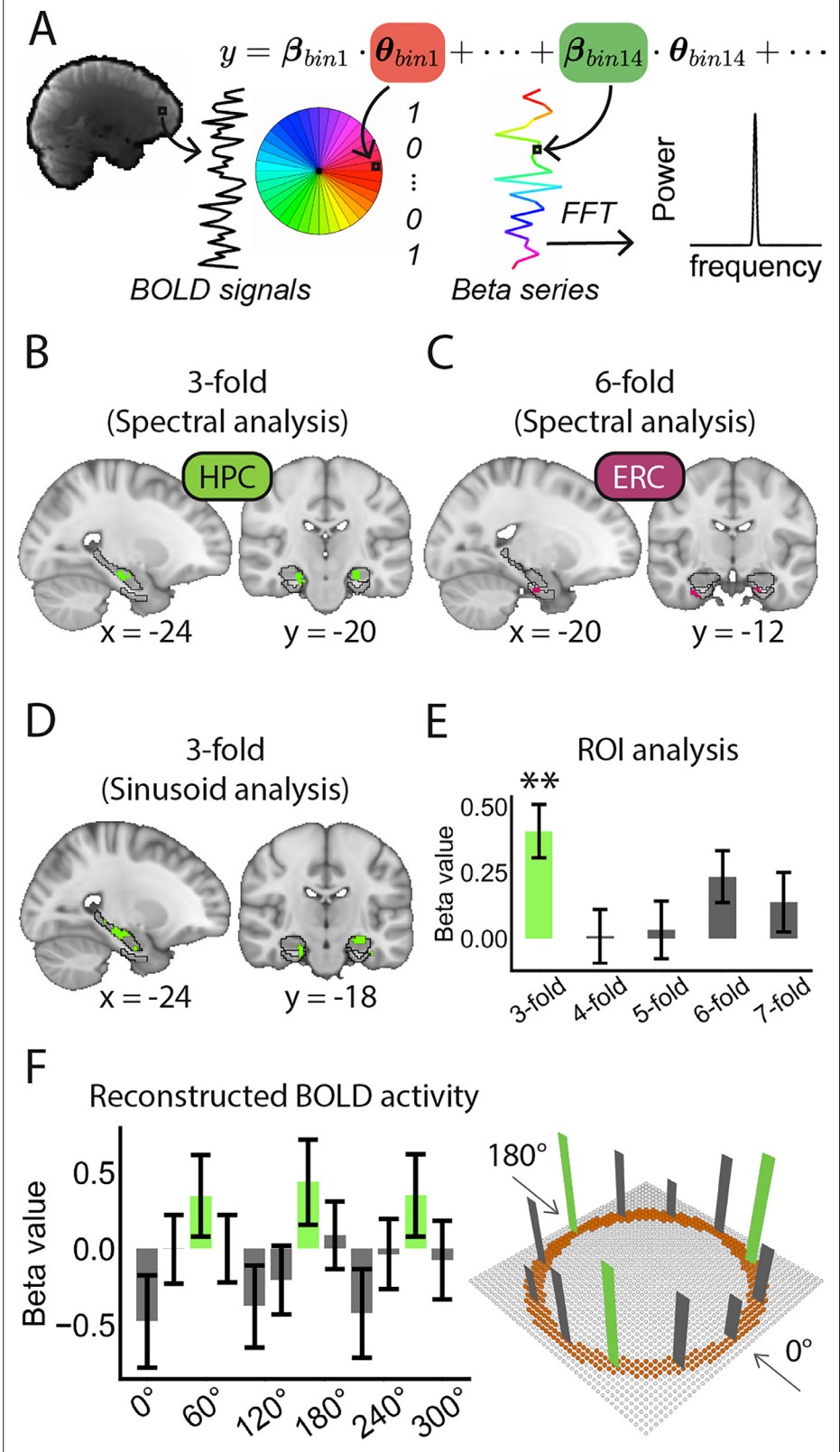

**Figure 3.** Threefold periodicity in the HPC. (**A**) Schematic illustration of the spectral analysis procedure. (**B–C**) Voxel-based spectral analysis revealed significant threefold periodicity in the bilateral HPC and sixfold periodicity in the bilateral EC (initial threshold: p=0.05, two-tailed; Cluster-based SVC correction for multiple comparisons: p<0.05; For the HPC: Cohen's d=1.06; Peak MNI coordinate: −24, −20, −18; For the EC: Cohen's

*Figure 3 continued*

d=1.27; Peak MNI coordinate: −22, −14, −30). The black lines indicate the boundaries of the HPC and EC. (**D–E**) Threefold periodicity in the bilateral HPC identified using sinusoidal modulation (initial threshold: p=0.05, two-tailed; cluster-based SVC correction: p<0.05; Cohen's d=0.68; peak MNI coordinate: −24, −18, −12). ROI-based analysis using a functional mask of this cluster within the anatomical HPC confirmed the effect (t(32) = 3.94, p=0.002; Cohen's d=0.70). Volumetric results are displayed in radiological orientation; numbers below the brain slices indicate MNI coordinates. (**F**) Schematic illustrating the threefold directional tuning curve reconstructed from sinusoidal modulation (Left) and its representation in the 2D Greeble space (Right). The error bars indicate SEM.

The online version of this article includes the following source data and figure supplement(s) for figure 3:

**Source data 1.** Related to *Figure 3E*.

**Source data 2.** Related to *Figure 3F*.

**Figure supplement 1.** Threefold periodicity of the HPC examined using 20° bin.

**Figure supplement 2.** Three and sixfold periodicity of participant groups revealed by spectral analysis.

**Figure supplement 3.** Whole brain representations of six and threefold periodicity revealed by spectral analysis.

First, sinusoidal modulation confirmed significant threefold clusters in the bilateral HPC (*Figure 3D*; initial threshold: p=0.05, two-tailed; Cluster-based SVC-corrected for multiple comparisons: p<0.05; Cohen's d=0.68; Peak MNI coordinate: −24, −18, −12). ROI analysis, using a functional mask of the HPC identified in the spectral analysis and further restricted within the anatomical HPC, indicated that HPC activity selectively fluctuated at threefold periodicity (*Figure 3E*; t(32) = 3.94, p=0.002, corrected for multiple comparisons across rotational symmetries; Cohen's d=0.70), whereas no significant activity periodicity was observed at other spatial fold (fourfold periodicity: t(32) = 0.09, p=0.93, Cohen's d=0.02; fivefold periodicity: t(32) = 0.31, p=0.76, Cohen's d=0.05; sixfold periodicity: t(32) = 2.36, p=0.12, Cohen's d=0.42; sevenfold periodicity: t(32) = 1.21, p=0.24, Cohen's d=0.21). The reconstructed BOLD signals, direction-calibrated for each participant based on their spatial phase estimated from sinusoidal modulation, showed a significant threefold periodic pattern in the bilateral HPC (*Figure 3F*, left; p<0.05, corrected for multiple comparisons). In sum, both the spectral and sinusoidal modulation confirmed a threefold periodic activity in the HPC across the conceptual directions within the 2D Greeble space (*Figure 3F*, right), which was distinct from the sixfold periodicity observed in the EC.

Second, the threefold HPC periodicity was not affected by the precision of directional sampling, as consistent findings were observed when 20° bins were applied (*Figure 3—figure supplement 1*; initial threshold: p=0.05, two-tailed; Cluster-based SVC corrected for multiple comparisons: p<0.05; Cohen's d=1.18; Peak MNI coordinate: 22, −24, −14). Third, we independently analyzed HPC and EC periodicity using spectral analysis for each of the three site-dependent experimental groups (see Materials and methods for details). These analyses showed reliable activity periodicities in both brain areas across experimental groups (*Figure 3—figure supplement 2*; initial threshold: p=0.05, two-tailed. Cluster-based SVC corrected for multiple comparisons: p<0.05). Fourth, we further examined the whole brain representation of both three and sixfold activity periodicity, respectively (*Figure 3—figure supplement 3*). The threefold periodicity revealed significant involvement of the medial prefrontal cortex (mPFC), precuneus (PCu), and parietal cortex (PC; initial threshold: p=0.05, two-tailed. Whole brain correction for multiple comparisons: p<0.05), suggesting engagement of the default mode network (DMN). In contrast, the sixfold periodicity highlighted the Salience network, including the anterior cingulate cortex (ACC) and insular cortex (INS), in addition to the EC (initial threshold: p=0.05, two-tailed; Whole brain cluster-based correction for multiple comparisons: p<0.05). These results demonstrated distinct functional networks involved in integrating spatial metric (sixfold periodicity) and in judging self-motion within conceptual space (threefold periodicity).

## Phase synchronization between the HPC and EC activity

To examine whether the spatial phase structure in one region could predict that in another, we tested whether the orientations of the sixfold EC and threefold HPC periodic activities, estimated from odd-numbered sessions using sinusoidal modulation with rotationally symmetric parameters, were correlated across participants. A cross-participant circular–circular correlation was conducted between the spatial phases of the two areas to quantify the spatial correspondence of their activity

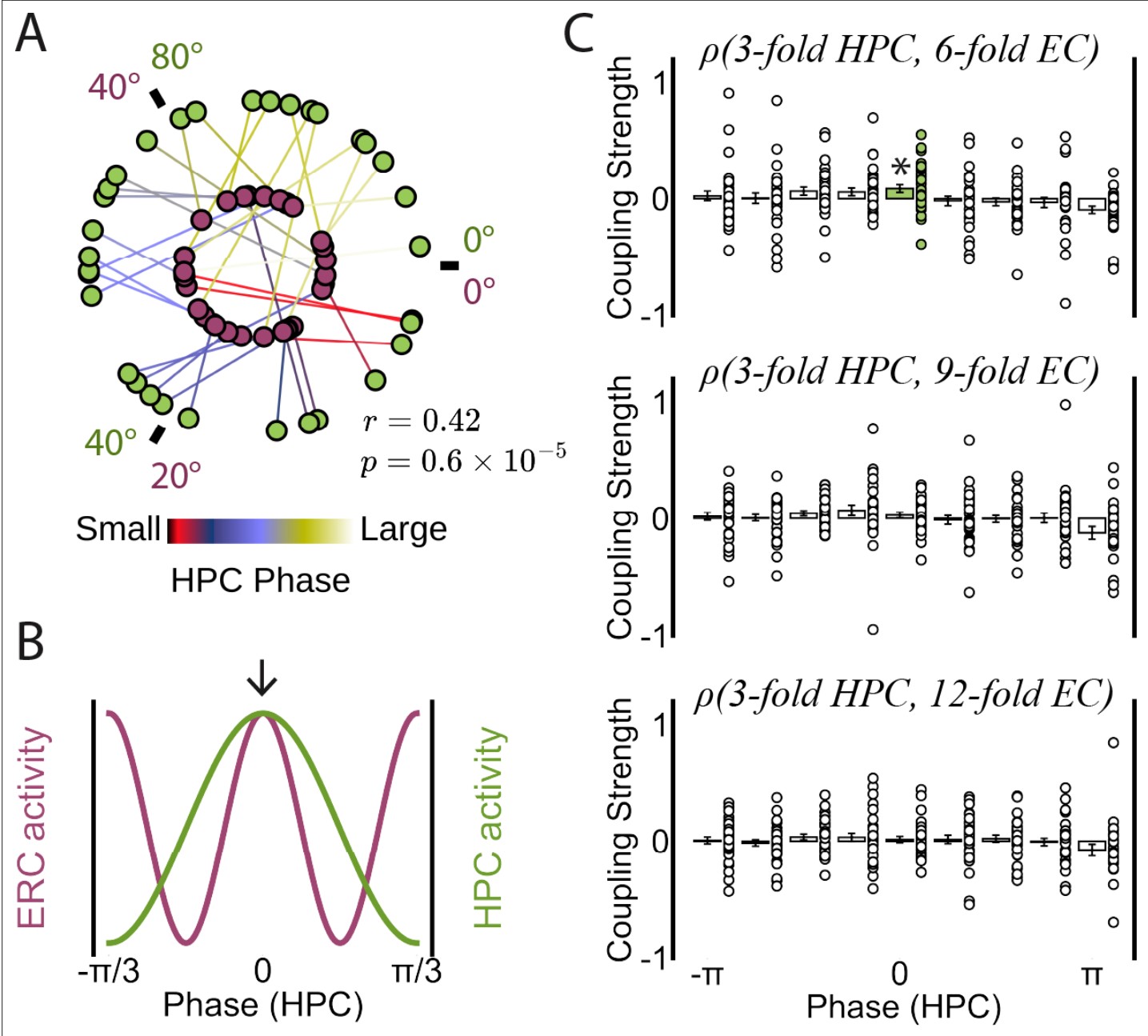

**Figure 4.** Phase synchronization between the HPC and EC activity in the directional space. (**A**) Cross-participant circular–circular correlation analysis revealed a significant coupling between the EC and HPC phases ($r=0.42$, $p<0.001$). The green (outer ring) and purple (inner ring) dots denote HPC and EC phases, respectively. Each line connecting the EC and HPC phases represents one participant, with its color indicating the HPC phase (0–120°). (**B**) Schematic illustration of a hypothetical peak-overlapping pattern (blue ellipse) between the HPC and EC activity in corresponding to spatial phase (one-cycle). (**C**) Amplitude-phase modulation analysis revealed significant coupling between the threefold HPC activity and sixfold EC activity in the bin centered at phase 0 ($t(32) = 2.57$, $p=0.02$, Bonferroni-corrected across tests; Cohen's d=0.45). The coupling strength was computed as the difference between the observed modulation index ($M$) and the mean surrogate modulation index ($M'$). No significant coupling was observed in the control analyses testing phase synchronization between the threefold HPC and the ninefold or twelvefold EC periodicities ($p>0.05$). The error bars indicate SEM.

The online version of this article includes the following source data for figure 4:

**Source data 1.** Related to *Figure 4A*.

**Source data 2.** Related to *Figure 4C*.

patterns (EC: purple dots; HPC: green dots; *Jammalamadaka and Sengupta, 2001*). The analysis revealed a significant circular correlation (*Figure 4A*; r=0.42, p<0.001), as reflected by the continuous color progression across the participants (i.e. the colored lines connecting each pair of the EC and HPC dots in *Figure 4A*), suggesting that participants with smaller hippocampal phases (green, outer ring) tended to have smaller entorhinal phases (purple, inner ring), and vice versa.

In addition to the across-participant phase correlation, we further examined the spatial alignment between the sixfold EC and threefold HPC activity patterns. Given that the spatial phase of the HPC is hypothesized to depend on EC projections, particularly along the three primary axes of the hexagonal code, we examined whether the periodic activities of the EC and HPC were spatially peak-aligned. Notably, unlike previous studies that focused on temporal coherence of neural oscillations (*Buzsaki, 2006*; *Maris et al., 2011*; *Friese et al., 2013*), our analysis focused on periodic coupling between brain areas in the directional space. To test spatial peak alignment between EC and HPC, a cross-frequency spatial coupling analysis (adapted from the amplitude–phase coupling framework; *Canolty et al., 2006*) was employed to identify at which HPC phase the EC exhibited maximal amplitude modulation. If the activities of both areas were peak-aligned (i.e. no peak offset), a strong coupling at phase 0 of the HPC would be expected as shown by the one-cycle-based schema in *Figure 4B*. In doing so, the instantaneous phase of the HPC and the amplitude envelope of the EC were extracted from the reconstructed activity using the Hilbert transform (see methods for details). HPC phases were classified into nine bins, and the modulation index (M), quantifying the scalar coupling strength between EC amplitude and HPC phase, was computed within each bin. As a result, significant coupling was observed in the bin centered at phase 0 of the HPC (*Figure 4C*; t(32) = 2.57, p=0.02, Bonferroni-corrected across tests; Cohen's d=0.45). In contrast, no significant coupling was found in other bins (p>0.05). To rule out the possibility that the observed coupling was driven by a potential harmonic (integer multiple) relationship between the threefold and sixfold periodicities, we additionally conducted control analyses using ninefold and twelvefold EC components. However, no significant coupling was observed in these controls (*Figure 4C*; p>0.05). Together, these results confirmed selective alignments of spatial peaks between the sixfold EC and threefold HPC periodicity in the conceptual direction domain.

## Threefold periodicity in human behavior

Considering the reciprocal connectivity within the medial temporal lobe (MTL), where the EC and HPC reside, and the parietal cortex implicated in visuospatial perception and action, together with the observed threefold periodicity within the DMN (including the PC and PCu; *Figure 3—figure supplement 3*), we hypothesized that the threefold periodic representations of path directions extend beyond the MTL to the egocentric cortical areas, such as the PC, thereby influencing participants' visuospatial task performance (*Figure 5A*). To test this hypothesis, we assessed participants' behavioral performance using two metrics: (1) path length and (2) the deviation (i.e. error size) between the actual ending location and the goal location per trial, since participants rarely stopped exactly at the goal. This procedure yielded, for each participant, a performance vector in the directional space (*Figure 5B*), with superior performance indexed by a composite score reflecting shorter path and smaller error size.

Using spectral analysis, we identified a significant periodicity in participants' behavioral performance, with spectral power peaking at a threefold symmetry (*Figure 5C*; p<0.05, permutation corrected for multiple comparisons), demonstrating that participants' visuospatial perception performance fluctuated as a function of path directions. In contrast, no significant spectral power was observed at other spatial folds, including sixfold behavioral periodicity (p>0.05). As a control, we also examined behavioral periodicity along each individual metric, path length, and error size. A significant threefold periodicity was observed for path length (*Figure 5—figure supplement 1*; p<0.05, permutation-corrected for multiple comparisons), whereas no significant modulation was found for error size, likely reflecting a residual fourfold bias inherent to the four cardinal-direction movement structure. To further investigate the causal relationship in brain-behavior coupling, we calculated the phase-lag index between the threefold behavioral periodicity and HPC activity for each participant (*Stam et al., 2007*). This analysis revealed a significantly higher phase-locking value (PLV) compared to surrogate dataset (*Figure 5D*; t(32) = 8.10, p<0.001; Cohen's d=1.14), confirming phase coupling between the threefold HPC activity and the threefold behavioral performance. Contrary to the commonly held assumption of isotropic task performance across path directions, these results demonstrated a biased

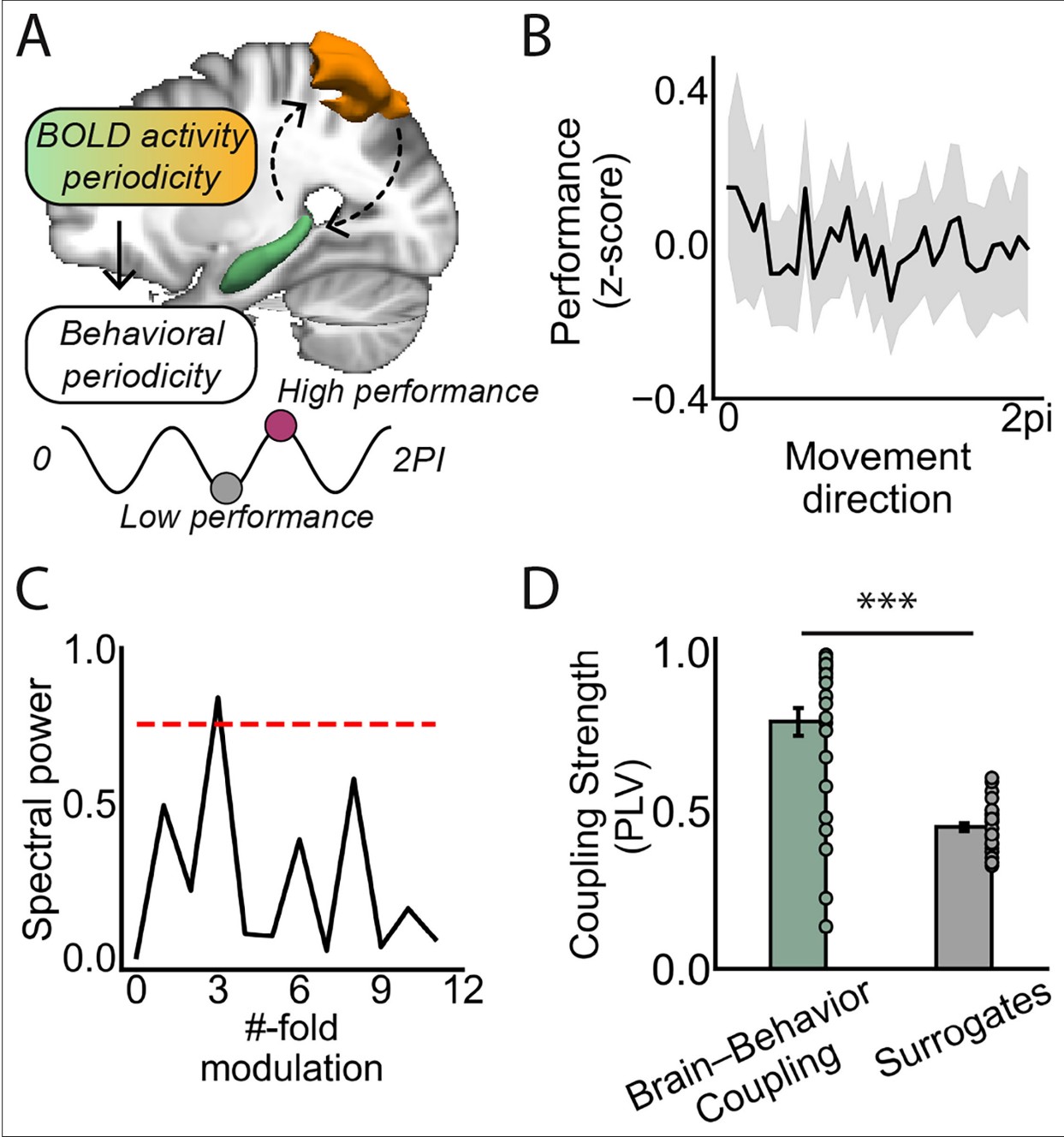

**Figure 5.** Threefold periodicity in behavioral performance. (**A**) Schematic illustration of visuospatial task performance potentially inheriting HPC's threefold periodicity. (**B**) Participants' behavioral performance, measured by a composite index of path length and deviation from the goal to ending locations, fluctuated as a function of path directions. The shaded area denotes SE. (**C**) Spectral analysis revealed significant power at the threefold of participants' behavioral performance (p<0.05, corrected for multiple comparisons). The red dashed line represents the baseline derived from permutation. (**D**) Significantly higher phase-locking values were observed between participants' behavioral performance and HPC activity compared to surrogate dataset (t(32) = 8.10, p<0.001; Cohen's d=1.14). The error bars indicate SEM.

The online version of this article includes the following source data and figure supplement(s) for figure 5:

**Source data 1.** Related to *Figure 5B*.

**Source data 2.** Related to *Figure 5C*.

**Source data 3.** Related to *Figure 5D*.

**Figure supplement 1.** Spectral analysis of human behavior.

behavioral pattern influenced by HPC and EC activity periodicity, with superior navigational performance (i.e. shorter path length and smaller error size) observed when path directions aligned with grid axes compared to when they were misaligned.

## The 'EC-HPC PhaseSync' model

The above empirical results elucidate that the vectorial representations of the HPC, characterized by threefold activity periodicity, were spatially coherent with EC population activity across the directional space. To mechanistically explain how the sixfold EC code gives rise to a threefold HPC representation, we constructed a cognitive model, termed the 'EC-HPC PhaseSync' model (*Figure 6*), which illustrates a possible computational mechanism of the EC-HPC circuit by simulating how the periodic representations emerged in the HPC influences spatial movements toward the center of Greeble space, as well as qualifying the reliability of the threefold periodic performance, particularly when goal locations are randomly distributed across the conceptual space.

The EC grid code $G$ was simulated using a cosine grating model (*O'Keefe and Burgess, 2005*; *Burgess et al., 2007*; *Blair et al., 2007*; *Bush and Schmidt-Hieber, 2018*; see methods for details). A population of 45-by-45 grid cells was generated to tessellate the space (*Figure 6A*, left), with spatial orientations kept constant while spatial phases varied across locations. The population activity in the EC, associated with movements from a start location, was termed the path code $V$, simulated by summing the grid codes across locations along the path (*Figure 6A*, right). Notably, the path code $V$ is exactly identical for path direction $\phi$ and $\phi + 180°$, naturally reflecting the spatial orientation $\psi \in [0, \pi)$ across the space.

The downstream HPC activity was simulated using a two-dimensional $\delta$ vector to represent spatial orientation. Each $\delta$ value, encoded by a single HPC neuron, was obtained by linearly summing EC activity across locations along the path code $V$ (*Figure 6B*, black dots). The spatial periodicity and spatial phase of $\delta$ depend on the number of primary axes and orientation of grid codes, while the magnitude of $\delta$ indicates the degree of alignment between the path orientation $\psi$ and the grid axes. Therefore, the threefold structure of $\delta$, projected from the EC grid cells, initially represents environmental orientations when self-location is not integrated. When self-location is anchored into space, this threefold $\delta$ structure embeds allocentric directions so that the preferred directions can be distinguished. The amplitude of $\delta$ reflects the directional modulation strength and the phase specifies the angular offset of the threefold pattern. The goal-directed vector representation $C$ in the HPC was next simulated by integrating the $\delta$ vector with a Gaussian-based distance vector centered at the goal location, thereby driving movements toward the goal (*Figure 6C*).

During simulations, 100 goal locations were randomly generated within the space, each paired with 120 starting locations, arranged in a ring-shaped pattern near the boundary. At each simulation step, a vector of HPC activity was extracted from the locations surrounding the current location, and the next movement was determined by the strongest activity of adjacent locations, following winner-take-all dynamics. The step size, initialized as 1 pixel, was dynamically adjusted using a stepwise approach, with a maximum limit of 5 pixels; for example, the step size increased by 1 pixel if no stronger activity was found among the surrounding locations compared to the activity of self-location. Model performance was defined by the path length, calculated as the sum of Euclidean distances between consecutive locations, with superior performance corresponding to shorter path length. The model performance of all directions was sorted in ascending order before spectral analysis, and the mean spectral powers, calculated across 100 navigation tasks, was compared with a permutation-based significant threshold. The simulations revealed a significant threefold periodicity in path length across 100 navigation tasks (*Figure 6D*; p<0.05, corrected for multiple comparisons). No other periodicities reached significance (p>0.05). These results demonstrate that projections from grid cell populations can reproduce the threefold behavioral periodicity observed in humans during goal-directed navigation.

## Discussion

Spatial navigation, the cognitive process of retrieving spatial relationships from one moment to the next in both conceptual (*Epstein et al., 2017*; *Behrens et al., 2018*; *Bottini and Doeller, 2020*; *Park et al., 2020*) and physical spaces (*Tolman, 1948*; *Sargolini et al., 2006*; *Gardner et al., 2022*; *Ormond and O'Keefe, 2022*), relies on the EC-HPC circuit. These two areas are known to have

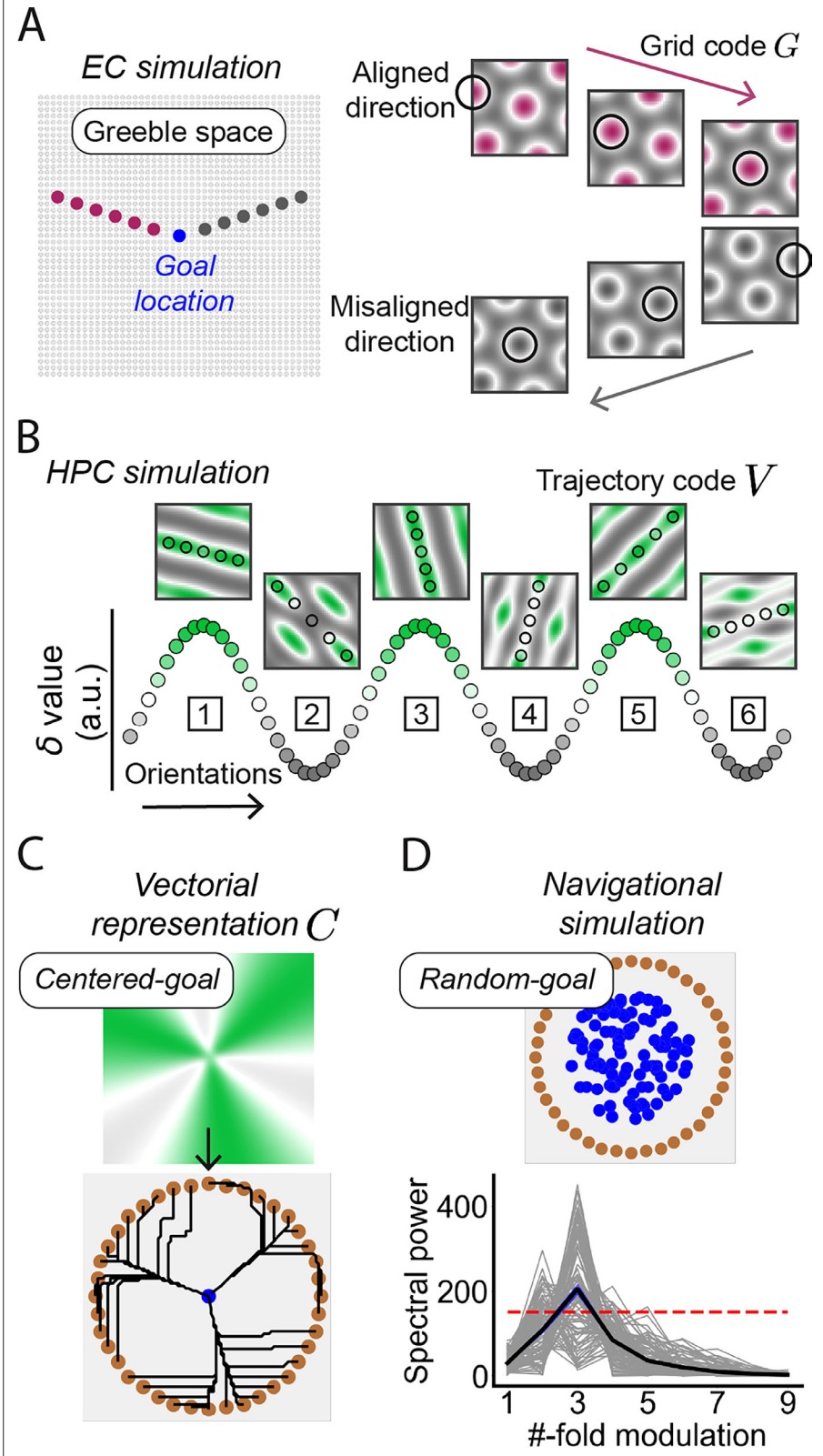

**Figure 6.** The EC-HPC PhaseSync model. (**A**) Schematic illustrating the population activity of grid cells during mental planning. The simulated grid cell population was activated by visiting discrete locations (the black circles in the right panel) in the Greeble space along directions either aligned (purple) or misaligned (gray) relative to the grid axes. (**B**) The threefold periodicity of the path code $V$ represented in the HPC. A path code $V$ is symmetrical

*Figure 6 continued*

for path direction $\phi$ and $\phi$ +180°, representing a unique spatial orientation $\psi$ ranging from 0° to 180°. The $\delta$ value (y-axis) indicates the degree to which the spatial orientation $\psi$ of a path code $V$ aligns with the grid axes, with larger $\delta$ values (e.g. the spatial orientation [1, 3, and 5]) indicating a perfect match between $\phi$ and grid axes. (**C**) Simulated vectorial representation of the HPC for centered-goal-based navigation. The threefold periodicity is driven by vectorial gradients inherited from $\delta$. (**D**) Random-goal-based navigational simulation. Significant spectral power of model performance was observed at threefold across 100 randomly selected goal locations (the blue dots and gray lines; p<0.05, corrected for multiple comparisons). The red dashed line indicates the significance threshold derived from permutation. The blue-shaded areas denote the standard error. Gray lines represent the spectral powers of goal-dependent simulations.

The online version of this article includes the following source data for figure 6:

**Source data 1.** Related to *Figure 6C*.

**Source data 2.** Related to *Figure 6D*.

distinct functions. The EC encodes continuous space with a metric representation (*Hafting et al., 2005*), while the HPC represents localized spatial locations (*O'Keefe and Dostrovsky, 1971*). A longstanding puzzle is how they collaborate to encode spatial navigation. In the present study, we modulated the EC and HPC activity using conceptual directions and identified sixfold periodicity from EC activity and threefold periodicity from HPC activity. The periodic activities of both areas were synchronized and peak-aligned. Additionally, threefold periodicity was also observed in participants' behavioral performance, which was synchronized with the HPC activity, suggesting a direct influence of periodic brain activity on visuospatial perception. Our findings provide a potential explanation for the observation of the hippocampal vector field (*Sarel et al., 2017*; *Ormond and O'Keefe, 2022*; *Muhle-Karbe et al., 2023*).

In addition to the HPC and EC, whole-brain spectral analysis revealed involvement of the DMN and salience network for the three and sixfold periodicity, respectively, suggesting that concept-dependent periodic neural activity is supported by distinct networks and extends beyond the EC–HPC circuit. These results do not imply that the grid cell's hexagonal code is the origin of the spatially selective periodic activity observed in these brain networks. Rather, these large-scale activity dynamics more likely reflect hierarchical computations across the brain. In upstream visual areas, neurons encode position-specific features and provide egocentric cues when an allocentric map is unavailable, as seen in the feature-selective neurons in V1 that represent locations across visual space, as described by *Hubel and Wiesel, 1959*; *Hubel and Wiesel, 1962*. Moreover, spatial attention has been shown to operate through rhythmic sampling in the temporal domain. Neural activity in the frontoparietal attention networks oscillates at 3–8 Hz and predicts trial-by-trial performance in spatial discrimination tasks (*Fiebelkorn et al., 2018*). This finding parallels our observed threefold behavioral periodicity, indexed by task performance, suggesting that neural representation in both the temporal and spatial domains may rely on periodic computations, potentially reflecting a shared mechanism by which the brain organizes perception and guides goal-directed behavior.

The EC–HPC PhaseSync model demonstrates how a vectorial representation may emerge in the HPC from the projections of populations of periodic grid codes in the EC. The model was motivated by two observations. First, the EC intrinsically serves as the major source of hippocampal input (*Witter and Amaral, 1991*; *van Groen et al., 2003*; *Garcia and Buffalo, 2020*), and grid codes exhibit nearly invariant spatial orientations (*Hafting et al., 2005*; *Gardner et al., 2022*). Second, mental planning, characterized by 'forward replay' (*Dragoi and Tonegawa, 2011*; *Pfeiffer, 2020*), has the capacity to activate populations of grid cells that represent sequential experiences in the absence of actual physical movement (*Nyberg et al., 2022*). We hypothesize that an integrated path code of sequential experiences may eventually be generated in the HPC, providing a vectorial gradient toward the goal location. The path code exhibits regular, vector-like representations when the path direction aligns with the orientations of grid axes and becomes irregular when they misalign. This explanation is consistent with the band-like representations observed in the dorsomedial EC (*Krupic et al., 2012*) and the irregular activity fields of trace cells in the HPC (*Poulter et al., 2021*).

In sum, our study identifies a threefold hippocampal code that complements the sixfold EC metrics, potentially supporting flexible navigation. Extending previous cognitive models based on grid- and place-cell simulations of navigation (*Bicanski and Burgess, 2019*; *Whittington et al., 2020*; *Edvardsen*

*et al., 2020*), our findings highlight a potential hierarchical periodic architecture of neurons within the MTL, particularly in representing spatial directions. Moreover, considering that the task space was simplified to an idealized square layout, in more complex, real-world environments, hippocampal threefold periodicity may interact with other spatial variables such as distance, movement speed, and environmental boundaries. Future studies should aim to characterize how the periodic activity of the EC–HPC circuit interacts with the external environment to support real-world navigation.

## Materials and methods

### Participants

Thirty-three right-handed university students with normal or corrected-to-normal vision participated in the experiment. Participants were recruited in three separate batches: 10 from Tsinghua University (Group 1: mean age = 23.9 [SD = 3.51], 5 females, 5 males), 10 from Peking University (Group 2: mean age = 22.10 [SD = 2.23], 5 females, 5 males), and 13 from Tsinghua University (Group 3: mean age = 20.85 [SD = 3.08], 2 females, 11 males). All participants had no history of psychiatric or neurological disorders and provided written informed consent prior to testing, including consent for the use and publication of anonymized data. The study protocol was approved by the Research Ethics Committee of the Faculty of Psychology, Beijing Normal University (approval number: 202003180020).

### Greeble space

The Greebles (*Gauthier and Tarr, 1997*) were generated based on the modified codes of the TarrLab stimulus datasets (http://www.tarrlab.org/) using Autodesk 3ds Max (version 2022, http://www.autodesk.com). A Greeble's identity was defined by the lengths of two distinct features, represented by two nonsense words, 'Loogit' and 'Vacso' (*Figure 1A*). These nonsense names were arbitrarily assigned and carried no semantic relevance to the task. The generated Greebles were arranged in a 45-by-45 matrix, forming a two-dimensional space, with each Greeble representing a 'location'. The feature 'Loogit' corresponded to the y-axis, while the feature 'Vacso' represented the x-axis (*Figure 1C*). The prototype Greeble, positioned at the center of the Greeble space (*Figure 1C*; blue dot), served as the target (i.e. 'goal location'), with Loogit and Vacso having equal lengths. The length ratios of Greebles' feature relative to the Greeble prototype ranged from 0.12 to 1.88, with a step size of 0.04, providing a smooth morphing experience as participants adjusted the lengths of the Greeble features.

### Experimental design

The object matching task was programmed using Pygame (version 2.0, https://www.pygame.org/; Python version 3.9). Each trial began with a 2.0 s fixation screen, followed by a pseudo-randomly selected Greeble variant presented at the center of the screen (*Figure 1B*). These variants were sampled from those near the boundary of the feature space (*Figure 1C*; orange dots). Participants were instructed to adjust the features of the Greeble variant by pressing buttons to match the Greeble prototype as quickly as possible. Two response boxes enabled participants to stretch or shrink the two features. These boxes were counterbalanced between the left and right hands of participants. To mitigate the impact of learning effects on BOLD signals, participants completed a training task one day prior to MRI scanning. On the training day, participants completed the task in a self-paced manner. They first familiarized themselves with the task and then performed at least 10 practice trials before the main training session, freely adjusting Greeble features and pressing the return key to end each trial. Feedback was provided during the first 7 sessions (e.g. 'Loogit: 5 percent longer than the target'), while no feedback was given in the final session. Day 2's MRI experiment procedure mirrored day 1, except that (1) participants were required to complete a trial within 10 s. A timer was presented on top of the screen (*Figure 1B*), and (2) no feedback was provided for all experimental sessions. Based on the present design, the 'path' from a Greeble variant towards the prototype can be interpreted as forming a conceptual direction. To ensure an even distribution of conceptual directions, we arbitrarily defined the 'movement' from the East as 0°. For each 30° bin, 24 Greeble variants were pseudo-randomly selected from the boundary of the feature space with an angular precision of 1.25°, resulting in a total of 288 Greeble variants distributed in eight experimental sessions. Each session contained 36 object matching trials and 4 lure trials. The lure trial presented a blank screen after the

fixation for 10 s. The MRI experiment lasted approximately 1 hr. Participants were incentivized to perform accurately with exponentially increasing monetary rewards based on their performance. After the scan, participants were debriefed, shown their performance results, and asked to discuss their subjective feelings as well as any perceived reasons for any performance issues. Importantly, participants were unaware of the existence of the Greeble space or the spatial features such as the location and direction during the task.

## Behavioral performance

The behavioral performance score was calculated for each trial as $T - T' + E/0.04$, where $T$ and $T'$ denote the participants' actual and the objectively optimal path length, respectively. The term $E/0.04$ quantified the error size, the Euclidean distance (in pixels) between the participants' ending locations (final adjusted Greeble) and the goal location (Greeble prototype). The constant '0.04' represents the step size of the Greeble feature ratio and was used to convert the ratio-based error to pixel units. The actual path length $T$ was defined as the number of movement steps (in pixels) recorded from the participants' paths, whereas the optimal path length $T'$ was determined by the number of pixels in the shortest possible path between the Greeble variants and Greeble prototype. Subtracting $T'$ from $T$ thus normalized each participant's performance relative to the optimal path. The rationale for this correction was to eliminate the systematic difference in path length between cardinal (i.e. 'East', 'South', 'West', and 'North') and intercardinal (e.g. 'Southeast') directions, which arises from the restricted movements along the horizontal and vertical axis of the squared Greeble space. In this design, movements along cardinal directions inherently result in shorter paths, while those along intercardinal directions become longer. Although this normalization effectively reduced the directional bias in path length, it did not remove the residual fourfold bias in error size. Nevertheless, because path length and error size capture complementary aspects of task performance—navigational efficiency and positional accuracy—we incorporated both terms in quantifying behavioral performance. Superior performance was represented by shorter paths and smaller errors, corresponding to lower performance scores.

## fMRI data acquisition

Imaging data were collected using a 3T Siemens Prisma scanner, equipped with a 64- and 20-channel receiver head coil at Tsinghua University and Peking University Imaging Center, respectively. Functional data were acquired with a Multi-band Echo Planar Imaging (EPI) sequence with the following parameters: Acceleration Factor: 2, TR = 2000 ms, TE = 30 ms, matrix size = 112 × 112 × 62, flip angle = 90°, resolution = 2 × 2 × 2.3 mm$^3$, 62 slices with a slice thickness of 2 mm and a gap of 0.3 mm, in a transversal slice orientation. Additionally, high-resolution T1-weighted three-dimensional anatomical datasets were collected for registration purposes using MPRAGE sequences, detailed as follows: TR = 2530 ms, TE = 2.98 ms, matrix size = 448 x 512 x 192, flip angle = 7°, resolution = 0.5 × 0.5×1 mm$^3$, 192 slices with a slice thickness of 1 mm, in the sagittal slice orientation. At Tsinghua University, stimuli were presented through a Sinorad LCD projector (Shenzhen Sinorad Medical Electronics) onto a 33-inch rear-projection screen. At Peking University, stimuli presentation was managed using a visual/audio stimulation system (Shenzhen Sinorad SA-9939) and an MRI-compatible 40-inch LED liquid crystal display, custom-designed by Shenzhen Sinorad Medical Electronics Co., Ltd. The screen resolution for both sites was 1024x768. Participants viewed the stimuli through an angled mirror mounted on the head coil.

## fMRI data preprocessing

The BOLD signal series of each scanning session was preprocessed independently using the FSL FEAT toolbox from the FMRIB's Software Library (version 6.0, https://fsl.fmrib.ox.ac.uk/fsl/fslwiki; *Smith et al., 2004*; *Woolrich et al., 2009*; *Jenkinson et al., 2012*). For each scanning session, the BOLD signals were corrected for motion artifacts, slice time acquisition differences, geometrical distortion using fieldmaps, and were applied with a high-pass filter with a cutoff of 100 s. Spatial smoothing was performed using a Gaussian kernel with a full width at half maximum (FWHM) of 5 mm. For group-level analysis, the preprocessed BOLD signals were registered to each participant's high-resolution anatomical image and subsequently normalized to the standard MNI152 image using FSL FLIRT

(*Jenkinson and Smith, 2001*). During normalization, the functional voxels were resampled to the resolution of 2x2 x 2 mm.

## Sinusoidal modulation

A GLM with sinusoidal regressors was employed to investigate the periodic neural activity in the EC and HPC, following previously established protocols (*Doeller et al., 2010*; *Constantinescu et al., 2016*; *Bao et al., 2019*; *Wagner et al., 2023*; *Raithel et al., 2023*). First, the grid orientation in each voxel within the EC from each participant was calculated using half of the dataset (odd-numbered sessions: 1, 3, 5, and 7). Specifically, a GLM was created for each session to model the BOLD signals of the EC with two parametric modulators: $\sin(6\theta)$ and $\cos(6\theta)$, convolved with the Double-Gamma hemodynamic response function. $\theta$ represents the path directions, derived from drawing a line between the starting and the ending location, as each raw adjustment step was limited to four cardinal directions. Defining direction by start–end vectors thus allowed full 0–360° coverage, consistent with previous grid-like coding studies (e.g. *Constantinescu et al., 2016*). The factor '6' represents the rotationally symmetric sixfold neural activity. The factors '3', '4', '5', and '7', misaligned with the primary axes of grid cells, were used as control parameters. The estimated weights $\beta_{sine}$ and $\beta_{cosine}$ were then used to compute the grid orientation $\varphi$, ranging from 0° to 59°, where $\varphi = \left[\arctan\left(\frac{\beta_{sine}}{\beta_{cosine}}\right)\right]/6$. The $\varphi$ maps of each participant were averaged across the voxels of a hand-drawn bilateral EC mask and odd-numbered sessions to represent the grid orientation. Second, the neural representation of hexagonal activity, modulated by path directions in Greeble space, was examined by another GLM using the other half of the dataset (even-numbered sessions: 2, 4, 6, and 8), with path direction calibrated by the participant-specific grid orientation. The GLM was specified with a cosine parametric modulator, $\cos(6[\theta - \Phi])$. Third, the hexagonal activity maps were averaged across the even-numbered sessions for each participant before entering second-level analysis. The same procedure was applied to the bilateral HPC to test for threefold periodicity. Specifically, voxel-wise phase estimates were obtained from odd sessions using $\sin(3\theta)$ and $\cos(3\theta)$ regressors, and the participant-specific threefold phase was obtained by averaging voxel-wise estimates across the bilateral HPC, using a mask derived from the AAL template. In the even sessions, path directions were rotationally calibrated by subtracting this participant-specific phase, and a GLM with $\cos(3[\theta - \Phi])$ was fitted. The resulting maps of threefold periodic modulation were averaged across even sessions for each participant and then submitted to group-level analysis. To present the directional tuning curves of EC and HPC activity in the angular domain of path directions, we reconstructed the BOLD signals for each participant using parameters estimated from sinusoidal modulation. Functional masks derived from significant subregions identified in the voxel-wise analyses were employed to capture their tuning profiles with maximum sensitivity.

## Spectral analysis of MRI BOLD signals

Spectral analysis was employed to examine the spatial periodicity of BOLD signals. This procedure consisted of three stages: First-level analysis. A GLM was constructed to estimate direction-dependent activity maps. Participants' path directions were down-sampled into 36 bins of 10° each (e.g. path directions between 0° and 10° were assigned to the 0° bin, whereas those between 350° and 360° were assigned to the 350° bin). These bins were entered as regressors in the GLM, yielding activity estimates associated with each directional bin for every participant. Second-level analysis. The direction-dependent activity maps were sorted in ascending order (from 0° to 360°), detrended, and processed with a Hanning window. They were then transformed from the spatial to the frequency domain using FFT (implemented in the stats package of R, version 4.0; https://www.r-project.org/) (*Landau and Fries, 2012*). The absolute values of the complex FFT outputs were used to generate spectral magnitude maps for each participant, covering periodicities from zero to eighteenfold. Third-level analysis. For group inference, the 95th percentile (two-tailed) of the pseudo-t distribution, computed as the mean spectral magnitude divided by the standard error across participants, was used as the initial threshold for family-wise error correction in multiple comparisons.

## Phase–amplitude coupling

To quantify the spatial peak relationship between EC and HPC BOLD activity, we implemented a cross-frequency amplitude–phase coupling analysis in the directional space (*Canolty et al., 2006*). Rather than analyzing raw BOLD signals, we reconstructed sixfold EC activity and threefold HPC

activity in each voxel using sinusoidal modulation weights ($\beta_{sine}$ and $\beta_{cosine}$) estimated from the raw BOLD signals. Specifically, activity was modeled as $\beta_{cosine}\cos(k\theta) + \beta_{sine}\sin(k\theta)$, where $k$ denotes the rotational symmetry. This approach selectively captures the hypothesized spatial symmetries of neural activity (e.g. sixfold or threefold periodicity) as a function of movement direction. For this coupling analysis, we used participants' original movement directions (i.e. without applying orientation calibration). The reconstructed sixfold EC and threefold HPC activity were then converted into analytic representations using the Hilbert transform, yielding the instantaneous phase of the HPC ($\phi_{HPC}$) and the amplitude envelope of the EC ($A_{ERC}$). HPC phases were classified into nine bins. The composite analytic signal, defined as $z = A_{ERC}e^{i\phi_{HPC}}$, was used to compute the modulation index $M$ (*Canolty et al., 2006*), defined as the absolute value of the mean of z values, quantifying the scalar coupling strength between EC amplitude and HPC phase within each bin. A surrogate dataset, a null distribution of the modulation indices ($M'$), was generated by spatially offsetting the EC amplitude relative to the HPC phase across all possible spatial lags. The mean of this surrogate distribution was used as the baseline reference against which the observed coupling strength was compared.

## Anatomical masks

The mask of the bilateral EC was manually delineated on the MNI T1 brain with 1 mm resolution packaged by FSL, using established protocols (*Insausti et al., 1998*), and the delineation software ITK-SNAP (Version 3.8, https://www.itksnap.org/pmwiki/pmwiki.php). The EC mask was then resampled to a 2 mm resolution. The anatomical mask of the bilateral HPC and the MTL was derived from the AAL atlas, an automated anatomical parcellation of the spatially normalized single-subject high-resolution T1 volume provided by the Montreal Neurological Institute (Version: AAL 3v2; https://www.gin.cnrs.fr/en/tools/aal/; *Rolls et al., 2020*).

## Spectral analysis of human behavior

FFT analysis was employed to examine participants' behavioral periodicity, following the same procedure used to detect activity periodicity in the EC and HPC. For each participant, a performance vector was constructed from their paths. These vectors were down-sampled into 10° bins from the original 288 conceptual directions (from 0° to 360°), sorted in ascending order, detrended, and processed with a Hanning window. This procedure yielded a resampled vector of 36 data points, which was then entered into the FFT analysis. The two-tailed 95th percentile of the spectral magnitude distribution was used as the initial threshold for the family-wise error correction across multiple comparisons.

## The 'EC-HPC PhaseSync' model

The EC grid-cell spatial code $G$ was simulated using the cosine grating model (*O'Keefe and Burgess, 2005*; *Burgess et al., 2007*; *Blair et al., 2007*; *Bush and Schmidt-Hieber, 2018*) in Python (version 3.9). At each Greeble location $r = (x, y)$, three cosine gratings, oriented 60° apart, were generated by rotating around the anchor point $r$ according to the rotation vector $k$, and then linearly summed (*Equation 1*). The parameter $A$ represents the amplitude of the cosine gratings, $c = (c_x, c_y)$ denotes the spatial phase offset of the grating pattern, and $\omega$ indicates the scale ratio (i.e. grid modules). This procedure generated a 45-by-45 grid cell population, with a constant grid orientation maintained across the population.

$$G_r = \sum_{i=1}^{3} A\cos\left(\omega_i\left(k_{i,x}\left(r_x - c_x\right) + k_{i,y}\left(r_y - c_y\right)\right)\right) \tag{1}$$

The path code $V$ of grid cell population was defined as the linear summation of grid codes across locations along the path (*Equation 2*), where $r$ and $q$ denote the starting location (i.e. Greeble variant) and the ending location (Greeble prototype), and $r'$ indexes the intermediate locations along the path from $r$ to $q$.

$$V_r = \sum r' \in trajectory\,(r, q)\,G_{r'} \tag{2}$$

$V$ exhibited a 'planar wave' pattern (*Welday et al., 2011*; *Krupic et al., 2012*) when the allocentric direction $\phi_r$, defined by $\arctan2\left(\frac{q_y - r_y}{q_x - r_x}\right)$, aligned with the primary axes of grid cells (i.e. orientation

135°; aligned conditions). In contrast, when the allocentric direction $\phi_r$ deviated from the grid axes (i.e. orientation 246°; misaligned conditions), irregular activity patterns emerged from the path code $V$. Note that the path code $V$ is exactly identical between the allocentric direction $\phi$ and $\phi + 180°$. In other words, the Greeble space can be tessellated by a vector of orientations over $\psi \in [0, \pi)$ in the orientation domain (**Equation 3**).

$$\psi_r = \left(\text{arctan2}\left(q_y - r_y, q_x - r_x\right)\right) mod\, \pi \tag{3}$$

The neural representations of spatial orientation $\psi$ can be simplified as a two-dimensional vector $\delta$, where each $\delta_r$, representing activity in the 2D orientation space, was derived from the path code $V_r$ by linearly summing the activity of a set of locations $R$ along the path (**Equation 4**). The magnitude of $\delta_r$ indexed the degree of alignment between the orientation $\psi$ and the grid axes, with larger values indicating closer alignment. In the orientation domain, the $\delta$ vector was expected to exhibit a threefold periodicity across environmental orientations, characterized by a repeating 'aligned-deviated' pattern driven by the periodic structure of the path codes $V(\psi)$ projected from grid cell population, with the spatial phase inherited from grid orientation.

$$\delta_r = \sum_{r \in R_\psi} V_{r,\psi} \tag{4}$$

To construct the goal-directed vector representation $C$ of the HPC for driving movements from self-location towards goal-location, the allocentric directions $\Phi$ between the location vector $R$ and the goal location were encoded by the $\delta$ periodic structure, with the spatial fold and spatial phase denoted by $\delta_c$ and $\delta_\varepsilon$, respectively (**Equation 5**). Then, $C$ was the linearly summation between $\delta$ and a Gaussian-based distance term centered at the goal location $q$ (**Equation 5**). The distance term is invariant to spatial directions but encodes spatial proximity to the goal. The parameter $\sigma$ represents the spatial spread, defined by the radius of the Greeble space.

$$C\left(R\right) = \cos\left(\delta_c \times \Phi_R + \delta_\varepsilon\right) + \exp\left(-\frac{\|R - q\|^2}{2\sigma^2}\right) \tag{5}$$

## Statistical analysis

Voxel-wise clusters from the sinusoidal modulation and spectral analyses were first identified using a cluster-defining threshold of p<0.05 (two-tailed t-test). Cluster significance was determined with cluster-based permutation (5000 iterations) using FSL randomise (v2.9; **Nichols and Holmes, 2002**), with small-volume correction (SVC) applied within the bilateral MTL. Clusters exceeding the 95th percentile of the maximal suprathreshold cluster size were deemed significant. The directional tuning reliability was assessed via a label-shuffling permutation, in which directional labels were randomly permuted 5000 times to derive a null distribution of spectral power. The 95th percentile of surrogate power defined the uncorrected threshold, and the across-fold 95th percentile served as the FWE–corrected threshold. For the PPC (**Vinck et al., 2010**), significance was tested using 5000 permutations of uniformly distributed random phases (0–2π) to generate a null distribution for comparison with the observed PPC. For behavioral periodicity, both human and EC–HPC PhaseSync model data were evaluated against shuffled null distributions (5000 iterations). Multiple comparisons across folds (0–18) were controlled using the maximum-statistic approach. Finally, coherence analyses were performed for EC–HPC and behavior–HPC coupling. Direction-domain signals were band-pass filtered at the three and sixfold using a two-way least-squares FIR filter (eegfilt.m, EEGLAB; **Delorme and Makeig, 2004**). EC–HPC coherence was quantified using amplitude–phase coupling (**Canolty et al., 2006**), and behavior–HPC coherence was quantified using the phase-lag index (PLI; **Stam et al., 2007**). The relationship between EC and HPC phases was evaluated using the circular–circular correlation (**Jammalamadaka and Sengupta, 2001**) implemented in the CircStat MATLAB toolbox. For all coherence and phase-coupling analyses, statistical significance was assessed using a non-parametric permutation test. Surrogate datasets were generated by circularly shifting the signal series along the direction axis across all possible offsets (**Canolty et al., 2006**), thereby preserving the within-domain phase structure while disrupting consistent phase alignment between signals. Each surrogate dataset

underwent identical filtering and coherence computation to obtain a null distribution. The observed coherence strength was then compared with this distribution using paired t-tests across participants.

## Acknowledgements

We sincerely thank Yue Wu, Yuannan Li, and Ao Li for their thoughtful discussions that greatly contributed to our data analysis. We are also grateful to Russell Epstein for his insightful feedback on the fMRI analysis and computational modeling, and to Dr. Michael Tarr for generously sharing the code used for generating the Greeble stimuli. This work was supported by the High-Performance Computing Platform of Peking University and the Resnick High Performance Computing Center of California Institute of Technology. The work was supported by the following funding sources: National Key R&D Program of China 2020AAA0105200 (JL), China Postdoctoral Science Foundation 2022M710470 (BZ), Beijing Municipal Science & Technology Commission & Administrative Commission of Zhongguancun Science Park Z221100002722012 (JL), Tsinghua University Guoqiang Institute 2020GQG1016 (JL), and Beijing Academy of Artificial Intelligence (JL).

## Additional information

### Funding

| Funder | Grant reference number | Author |
| --- | --- | --- |
| National Key R&D Program of China | 2020AAA0105200 | Jia Liu |
| China Postdoctoral Science Foundation | 2022M710470 | Bo Zhang |
| Beijing Municipal Science and Technology Commission, Administrative Commission of Zhongguancun Science Park | Z221100002722012 | Jia Liu |
| Tsinghua University GuoQiang Institute | 2020GQG1016 | Jia Liu |
| Beijing Academy of Artificial Intelligence | | Jia Liu |

The funders had no role in study design, data collection and interpretation, or the decision to submit the work for publication.

### Author contributions

Bo Zhang, Data curation, Software, Formal analysis, Funding acquisition, Validation, Investigation, Visualization, Methodology, Writing – original draft, Project administration, Writing – review and editing; Xin Guan, Investigation; Dean Mobbs, Resources, Software, Writing – review and editing; Jia Liu, Conceptualization, Resources, Supervision, Funding acquisition, Writing – original draft, Project administration, Writing – review and editing

### Author ORCIDs

Bo Zhang https://orcid.org/0000-0002-0955-5585
Xin Guan https://orcid.org/0009-0004-3358-6279
Dean Mobbs https://orcid.org/0000-0003-1175-3772
Jia Liu https://orcid.org/0000-0003-0383-0934

### Ethics

The experimental protocol was reviewed and approved by the Research Ethics Committee of the Faculty of Psychology, Beijing Normal University (approval number: 202003180020). This approval covered all experimental procedures conducted at Peking University and Tsinghua University. All participants provided written informed consent prior to participation, including consent for the use

and publication of their anonymized data for research purposes. Participants were informed about the experimental procedures, potential risks, and their right to withdraw from the study at any time without penalty.

Reviewer #1 (Public review): https://doi.org/10.7554/eLife.107517.4.sa1
Reviewer #2 (Public review): https://doi.org/10.7554/eLife.107517.4.sa2
Author response https://doi.org/10.7554/eLife.107517.4.sa3

## Additional files

### Supplementary files
MDAR checklist

### Data availability
The object-matching task was programmed using Pygame (version 2.0). Analyses were carried out using custom scripts written in Python (version 3.9), MATLAB (version 2019b), R (version 4.0), and GNU Bash (version 3.2.57). Neuroimaging analyses were performed using FreeSurfer (version 7.1; https://surfer.nmr.mgh.harvard.edu/) and FSL (version 6.0; https://fsl.fmrib.ox.ac.uk/fsl/fslwiki). The Pygame and analysis code are publicly available at https://github.com/ZHANGneuro/Greeble_PyGame_fMRI_task (copy archived at *Zhang, 2026*). The MRI dataset, behavioral data, and manually delineated EC mask are available from the Science Data Bank at https://doi.org/10.57760/sciencedb.18351. The original code for the EC-HPC PhaseSync model is accessible at https://github.com/ZHANGneuro/The-E-H-PhaseSync-Model (copy archived at *Zhang, 2025*).

The following dataset was generated:

| Author(s) | Year | Dataset title | Dataset URL | Database and Identifier |
|---|---|---|---|---|
| Zhang B, Liu J | 2024 | An MRI dataset of human participants in Object Matching Task | https://doi.org/10.57760/sciencedb.18351 | Science Data Bank, 10.57760/sciencedb.18351 |

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
