## [Editor Report · eLife Assessment]

This study offers **important** insights into how entorhinal and hippocampal activity support human thinking in feature spaces. It replicates hexagonal symmetry in entorhinal cortex, reports a novel threefold symmetry in both behavior and hippocampal signals, and links these findings with a computational model. The task and analyses are sophisticated, and the results appear **convincing** and of broad interest to neuroscientists.

---

## [Referee Report · Reviewer #1 (Public review)]

Summary:

Zhang and colleagues examine neural representations underlying abstract navigation in entorhinal cortex (EC) and hippocampus (HC) using fMRI. This paper replicates a previously identified hexagonal modulation of abstract navigation vectors in abstract space in EC in a novel task involving navigating in a conceptual Greeble space. In HC, the authors identify a threefold signal of the navigation angle. They also use a novel analysis technique (spectral analysis) to look at spatial patterns in these two areas and identify phase coupling between HC and EC. Interestingly, the threefold pattern identified in the hippocampus explains quirks in participants' behavior where navigation performance follows a threefold periodicity. Finally, the authors propose a EC-HPC PhaseSync Model to understand how the EC and HC construct cognitive maps. The wide array and creativity of the techniques used is impressive but because of their unique nature, the paper would benefit from more details on how some of these techniques were implemented.

---

## [Referee Report · Reviewer #2 (Public review)]

The authors report results from behavioral data, fMRI recordings, and computer simulations during a conceptual navigation task. They report threefold symmetry in behavioral and simulated model performance, threefold symmetry in hippocampal activity, and sixfold symmetry in entorhinal activity (all as a function of movement directions in conceptual space). The analyses seem thoroughly done, and the results and simulations are very interesting.

[Editors' note: this version was assessed by the editors without consulting the reviewers further.]

---

## [Author Response]

The following is the authors’ response to the previous reviews

**Public Reviews:**

**Reviewer #1 (Public review):**
Summary:Zhang and colleagues examine neural representations underlying abstract navigation in entorhinal cortex (EC) and hippocampus (HC) using fMRI. This paper replicates a previously identified hexagonal modulation of abstract navigation vectors in abstract space in EC in a novel task involving navigating in a conceptual Greeble space. In HC, the authors identify a three-fold signal of the navigation angle. They also use a novel analysis technique (spectral analysis) to look at spatial patterns in these two areas and identify phase coupling between HC and EC. Interestingly, the three-fold pattern identified in the hippocampus explains quirks in participants' behavior where navigation performance follows a three-fold periodicity. Finally, the authors propose a EC-HPC PhaseSync Model to understand how the EC and HC construct cognitive maps. The wide array and creativity of the techniques used is impressive but because of their unique nature, the paper would benefit from more details on how some of these techniques were implemented.Comments on revisions:Most of my concerns were adequately addressed, and I believe the paper is greatly improved. I have two more points. I noticed that the legend for Figure 4 still refers to some components of the previous figure version, this should be updated to reflect the current version of the figure. I also think the paper would benefit from more details regarding some of the analyses.Specifically, the phase-amplitude coupling analysis should have a section in the methods which should be sure to clarify how the BOLD signals were reconstructed.(1)“…I noticed that the legend for Figure 4 still refers to some components of the previous figure version, this should be updated to reflect the current version of the figure…”.

Thank you for pointing this out. We have revised the legend of Figure 4 by removing the significance notation “***: p < 0.001”, which referred to elements from a previous version of the figure.

(2)“…I also think the paper would benefit from more details regarding some of the analyses. Specifically, the phase-amplitude coupling analysis should have a section in the methods which should be sure to clarify how the BOLD signals were reconstructed”.

We agree and appreciate the reviewer’s helpful suggestion. We have added a dedicated subsection entitled “Phase–amplitude coupling” to the Materials and Methods, in which we provide a detailed description of how the EC and HPC BOLD signals were reconstructed and how the coupling analysis was implemented. Correspondingly, we refined the description of this analysis in the Results section under “Phase synchronization between the HPC and EC activity”. The revised sections have been included below for your convenience.

Materials and Methods: Phase–amplitude coupling

To quantify the spatial peak relationship between EC and HPC BOLD activity, we implemented a cross-frequency amplitude–phase coupling analysis in the directional space (Canolty et al., 2006). Rather than analyzing raw BOLD signals, we reconstructed 6-fold EC activity and 3-fold HPC activity in each voxel using sinusoidal modulation weights (β_sine_ and β_cosine_) estimated from the raw BOLD signals. Specifically, activity was modeled as β_cosine_cos(kθ) + β_sine_sin(kθ), where k denotes the rotational symmetry. This approach selectively captures the hypothesized spatial symmetries of neural activity (e.g., 6-fold or 3-fold periodicity) as a function of movement direction. For this coupling analysis, we used participants’ original movement directions (i.e., without applying orientation calibration). The reconstructed 6-fold EC and 3-fold HPC activity were then converted into analytic representations using the Hilbert transform, yielding the instantaneous phase of the HPC (ϕ_HPC_) and the amplitude envelope of the EC (A_ERC_). HPC phases were classified into nine bins. The composite analytic signal, defined as z = A_ERC_e^iϕHPC^, was used to compute the modulation index M (Canolty et al., 2006), defined as the absolute value of the mean of z values, quantifying the scalar coupling strength between EC amplitude and HPC phase within each bin. A surrogate dataset, a null distribution of the modulation indices (M^-^), was generated by spatially offsetting the EC amplitude relative to the HPC phase across all possible spatial lags. The mean of this surrogate distribution was used as the baseline reference against which the observed coupling strength was compared.

Results: Phase synchronization between the HPC and EC activity

To examine whether the spatial phase structure in one region could predict that in another, we tested whether the orientations of the 6-fold EC and 3-fold HPC periodic activities, estimated from odd-numbered sessions using sinusoidal modulation with rotationally symmetric parameters, were correlated across participants. A cross-participant circular correlation was conducted between the spatial phases of the two areas to quantify the spatial correspondence of their activity patterns (EC: purple dots; HPC: green dots) (Jammalamadaka & Sengupta, 2001). The analysis revealed a significant circular correlation (Fig. 4a; r = 0.42, p < 0.001), as reflected by the continuous color progression across the participants (i.e., the colored lines connecting each pair of the EC and HPC dots in Fig. 4a), suggesting that participants with smaller hippocampal phases (green, outer ring) tended to have smaller entorhinal phases (purple, inner ring), and vice versa.

In addition to the across-participant phase correlation, we further examined the spatial alignment between the 6-fold EC and 3-fold HPC activity patterns. Given that the spatial phase of the HPC is hypothesized to depend on EC projections, particularly along the three primary axes of the hexagonal code, we examined whether the periodic activities of the EC and HPC were spatially peak-aligned. Notably, unlike previous studies that focused on temporal coherence of neural oscillations (Buzsaki, 2006; Maris et al., 2011; Friese et al., 2013), our analysis focused on periodic coupling between brain areas in the directional space. To test spatial peak alignment between EC and HPC, a cross-frequency spatial coupling analysis (adapted from the amplitude–phase coupling framework; Canolty et al., 2006) was employed to identify at which HPC phase the EC exhibited maximal amplitude modulation. If the activities of both areas were peak-aligned (i.e., no peak offset), a strong coupling at phase 0 of the HPC would be expected as shown by the one-cyclebased schema in Fig. 4b. In doing so, the instantaneous phase of the HPC and the amplitude envelope of the EC were extracted from the reconstructed activity using the Hilbert transform (see methods for details). HPC phases were classified into nine bins, and the modulation index (M), quantifying the scalar coupling strength between EC amplitude and HPC phase, was computed within each bin. As a result, significant coupling was observed in the bin centered at phase 0 of the HPC (Fig. 4c; t(32) = 2.57, p = 0.02, Bonferroni-corrected across tests; Cohen’s d = 0.45). In contrast, no significant coupling was found in other bins (p > 0.05). To rule out the possibility that the observed coupling was driven by a potential harmonic (integer multiple) relationship between the 3-fold and 6-fold periodicities, we additionally conducted control analyses using ninefold and twelvefold EC components. However, no significant coupling was observed in these controls (Fig. 4c; p > 0.05). Together, these results confirmed selective alignments of spatial peaks between the 6fold EC and 3-fold HPC periodicity in the conceptual direction domain.

**Reviewer #2 (Public review):**
The authors report results from behavioral data, fMRI recordings, and computer simulations during a conceptual navigation task. They report 3-fold symmetry in behavioral and simulated model performance, 3-fold symmetry in hippocampal activity, and 6-fold symmetry in entorhinal activity (all as a function of movement directions in conceptual space). The analyses seem thoroughly done, and the results and simulations are very interesting.

We thank the reviewer for the positive assessment of our work.

We thank both reviewers again for their constructive and insightful feedback, which has substantially strengthened the manuscript.